# Addition of Probiotics to Anti-Obesity Therapy by Percutaneous Electrical Stimulation of Dermatome T6. A Pilot Study

**DOI:** 10.3390/ijerph17197239

**Published:** 2020-10-03

**Authors:** Oscar Lorenzo, Marta Crespo-Yanguas, Tianyu Hang, Jairo Lumpuy-Castillo, Artur M. Hernández, Carolina Llavero, MLuisa García-Alonso, Jaime Ruiz-Tovar

**Affiliations:** 1Laboratory of Diabetes and Vascular Pathology, Instituto de Investigaciones Sanitarias-Fundación Jiménez Díaz, Universidad Autónoma, 28040 Madrid, Spain; marta.crespo@fjd.es (M.C.-Y.); taniahang9318@gmail.com (T.H.); jairo.lumpuy@estudiante.uam.es (J.L.-C.); mga@eaac.es (M.G.-A.); 2Spanish Biomedical Research Centre on Diabetes and Associated Metabolic Disorders (CIBERDEM) Network, 28040 Madrid, Spain; 3Department of Sport Sciences, Universidad Europea de Madrid, 28670 Villaviciosa de Odón-Madrid, Spain; ahernandez@urj.es; 4Obesity Unit, Clinica Garcilaso, 28010 Madrid, Spain; cllavero@urj.es (C.L.); jruiztovar@gmail.com (J.R.-T.); 5Department of Health Sciences, Universidad Rey Juan Carlos, 28933 Mostoles-Madrid, Spain

**Keywords:** obesity, percutaneous electrical stimulation, dermatome T6, microbiota

## Abstract

Obesity is becoming a pandemic and percutaneous electrical stimulation (PENS) of dermatome T6 has been demonstrated to reduce stomach motility and appetite, allowing greater weight loss than isolated hypocaloric diets. However, modulation of intestinal microbiota could improve this effect and control cardiovascular risk factors. Our objective was to test whether addition of probiotics could improve weight loss and cardiovascular risk factors in obese subjects after PENS and a hypocaloric diet. A pilot prospective study was performed in patients (*n* = 20) with a body mass index (BMI) > 30 kg/m^2^. Half of them underwent ten weeks of PENS in conjunction with a hypocaloric diet (PENS-Diet), and the other half was treated with a PENS-Diet plus multistrain probiotics (*L. plantarum LP115*, *B. brevis B3*, and *L. acidophilus LA14*) administration. Fecal samples were obtained before and after interventions. The weight loss and changes in blood pressure, glycemic and lipid profile, and in gut microbiota were investigated. Weight loss was significantly higher (16.2 vs. 11.1 kg, *p* = 0.022), whereas glycated hemoglobin and triglycerides were lower (−0.46 vs. −0.05%, *p* = 0.032, and −47.0 vs. −8.5 mg/dL, *p* = 0.002, respectively) in patients receiving PENS-Diet + probiotics compared with those with a PENS-Diet. Moreover, an enrichment of anti-obesogenic bacteria, including *Bifidobacterium spp, Akkermansia spp, Prevotella spp*, and the attenuation of the Firmicutes/Bacteroidetes ratio were noted in fecal samples after probiotics administration. In obese patients, the addition of probiotics to a PENS intervention under a hypocaloric diet could further improve weight loss and glycemic and lipid profile in parallel to the amelioration of gut dysbiosis.

## 1. Introduction

About a third of the population in developed countries is obese in some degree [1]. The WHO has proposed the classification of normal weight to be when the body mass index (BMI) ranges between 18.5 and 24.9 kg/m^2^, overweight (BMI 25–29.9 kg/m^2^), class I obesity (BMI 30–34.9 kg/m^2^), class II obesity (BMI 35–39.9 kg/m^2^), and class III obesity (BMI ≥ 40 kg/m^2^) [2]. Obesity itself is a health risk factor that influences the development and progression of various metabolic and cardiovascular diseases, such as dyslipidemia, type−2 diabetes mellitus (T2D), hypertension, and ischemic heart disease, thereby worsening the quality of life of patients and survival [3]. This non-communicable disease is also associated with low-grade, chronic systemic inflammation by dysregulation of adipokines and pro-inflammatory mediators (i.e., cytokines, chemokine), and subsequent alterations in the immune cell composition and distribution [4]. Obesity is a multifactorial disease and thus, therapeutic approaches should be diverse [5]. In this sense, dietary treatments associated with body exercise are primary anti-obesity approaches. Other strategies, such as behavior therapy, have been also considered [6]. However, long-term strategies with hypocaloric diets and physical exercise are not frequently attained, and psychological comorbidities may be chronic in these patients. In consequence, alternative procedures should be explored. In this regard, by percutaneous electrical neurostimulation (PENS) of the sensory nerve terminals located in dermatome T6, the gastric wall can be stimulated to produce distention in the fasting state, and to block contractions in the postprandial phase [7]. As a result, stomach emptying is slowed and early satiety is promoted. Moreover, the associated modulation of neuronal activities influences on appetite reduction [8]. Indeed, we previously demonstrated that PENS achieved a significantly greater weight loss than an isolated hypocaloric diet in patients with BMI > 30 kg/m^2^, and this effect could be due, at least in part, to ghrelin inhibition [9,10].

Importantly, pathogenesis of obesity has been linked with alterations in gut microorganisms. The intestinal microbiota is composed of tens of trillions of microorganisms, including at least 1000 different species of known bacteria, placed in the gut lumen or adhered to the mucus layer [11]. The five dominant bacterial phyla in the human gut are Firmicutes, Bacteroidetes, Actinobacteria, Proteobacteria, and Verrumicrobia [12]. The immunomodulatory bacteria are of great importance for local and systemic immunity, whereas the muconutritive microbiota are responsible of mucus layer formation, and the proteolytic bacteria have key metabolic functions in protein digestion [13]. Among other functions, microbiota participates in the metabolism of proteins, plant polyphenols, bile acids, and vitamins, and in the assimilation of non-absorbable carbohydrates by conversion into monosaccharides and short chain fatty acids (SCFA) and gases. However, though the intestinal microbiota is highly diverse in “healthy” individuals, those exhibiting adiposity, insulin resistance and/or dyslipidemia are characterized by low bacterial diversity [14]. Furthermore, obesity is associated with substantial changes in the composition and metabolic functions of bacteria, making an “obese microbiota”, which involves a greater extraction of nutrients from the diet [15]. Bacteroidetes prevalence is generally lower in obese people, in contrast with that of Firmicutes. However, the complexity of how the gut microbiome modulates obesity can be more than a simple disproportion between these commensal phyla [16,17]. In this line, different probiotics have demonstrated that they can balance microbiota bacteria and subsequently reduce body weight and metabolic and cardiovascular factors [18]. Thus, the aim of this study was to investigate the effect of probiotics on anti-obesity actions of PENS in conjunction with a hypocaloric diet.

## 2. Methodology

### 2.1. Subjects of Study

A pilot prospective study (NCT03872245) was completed in the Obesity Unit of Garcilaso Clinic (Madrid, Spain). The inclusion criteria were adult patients with a body mass index (BMI) > 30 kg/m^2^, with previous failure in dietary treatment. The exclusion criteria were (i) untreated endocrine diseases causing obesity, (ii) portable electric devices, (iii) diagnosis of previous cardiovascular events (acute myocardial infarction or coronary syndrome, heart failure) or cancer, and (iv) earlier treatment with hormone, prebiotics, probiotics, or nutritional supplements. In a previous study, we had observed that PENS of dermatome T6 (PENS) associated with the hypocaloric (1200 Kcal) diet produced a significantly greater weight loss (BMI = −5.1 kg/m^2^) than only PENS (BMI = −1.4 kg/m^2^) or the isolated hypocaloric diet (BMI = −2.0 kg/m^2^) [9]. Moreover, data from the literature have shown that single or multistrain probiotics alone produced minimal changes in body weight (BMI = −0.36 and −0.15 kg/m^2^, respectively) and in glycemic/lipidemic factors [19]. Therefore, we have now treated obese subjects, who previously were unsuccessfully treated only with the hypocaloric diet, with PENS with or without probiotics under the same diet in order to observe the potential differences in weight loss, associated cardiovascular factors (i.e., blood pressure, glycemia, and lipidemia), and microbiota. Thus, patients (*n* = 20) were randomized into two groups for anti-obesity interventions; PENS in conjunction with a hypocaloric diet (*n* = 10) (PENS-Diet), and the same strategy plus an administration of probiotics (*n* = 10) (PENS-Diet + probiotics). We followed a simple randomization using a random number table. All patients signed an informed consent for inclusion in the study and the use of clinical data for this research project. The Ethical Committee of Clinical Research (Medicine, Esthetic and longevity Foundation) approved this investigation (ref.: Garcilas-19-3; Feb 2019). The work was carried out in accordance with The Code of Ethics of the World Medical Association (Declaration of Helsinki). All participants finished the study.

### 2.2. Percutaneous Electrical Stimulation (PENS)

The PENS of dermatome T6 was performed as previously described [10] by using the Urgent PC 200 Neuromodulation System^®^ (Uroplasty, Minnetonka, MN, USA). Patients were placed in a supine position without anesthesia, and PENS was delivered by a needle electrode inserted in the left upper quadrant along the medio-clavicular line, at two centimeters below the ribcage, at a 90° angle towards the abdominal wall, and at 0.5–1 cm of depth. Successful insertion was confirmed by the feeling of electric movement at least 5 cm beyond the dermatome territory. The PENS was undertaken at a frequency of 20 Hz at the highest amplify (0–20 mA) without causing any pain. The participants underwent one 30-min session every week for ten consecutive weeks.

### 2.3. Hypocaloric Diet, Exercise, and Probiotics Administration

A 1200 Kcal/day diet was uniformly prescribed during PENS interventions in both groups of patients, as we previously published. The diet followed a Mediterranean style (carbohydrates 51%, proteins 23% and fat 26%) with a high intake of fruit and vegetables, a moderate intake of meats, and olive oil as the main source of fat [20]. Briefly, patients were recommended to take skimmed milk (200 mL) or natural yogurt and bread (200 g) as breakfast, 100 g of fruit (e.g., apple, pear) mid-morning, and 200 g of vegetables (e.g., spinach, lettuce, cauliflower) or pasta soup, fish (120 g) or chicken (100 g), and fruit (100 g), at lunch and dinner. Olive oil (30 cc) could be also taken as a complement, and skimmed milk (200 mL) with coffee or tea as a snack. A record of food intake was applied along the study. The intake of alcohol and nutritional supplements was not allowed during the study. Moreover, patients received instructions for regular exercise practice (1 h of brisk walking/day), following a counselling protocol against obesity in patients under 50 years [21]. Brisk walking consisted of a moderate-intensity exercise of walking to a minimum speed of 100 steps per minute (about 4.8 km/h). Since obese patients have many difficulties in adhering to nutritional advice and exercise recommendations, we did a weekly follow-up of food intake and exercise practice. Our dietician phoned all the patients to remind them of the need to stick to these recommendations. The dietician wrote down the daily intake of food and the time/speed of brisk walking. At the end of the study (10 weeks), the dietician confirmed a full adherence rate to the Mediterranean diet and daily exercise. In a previous work, we described a 98% and 94% diet compliance in patients undergoing PENS or PENS-Diet interventions, respectively [9]. The reduction of appetite induced by PENS and the short length of the study could facilitate this high adherence. Some patients additionally received two tablets per day of probiotics Adomelle^®^ (4th generation technology, Bromatech, Italy) during the ten weeks of treatment. The composition of Adomelle^®^ was *Lactobacillus plantarum LP115* (<1 × 10^9^ colony forming units; CFU)*, Bifidobacterium brevis B3* (<1 × 10^9^ CFU), and *Lactobacillus acidophilus LA14* (1 × 10^9^ CFU). The probiotics were given after meals with drinking water and did not alter the food intake. Participants were also compliant with probiotics intake.

### 2.4. Variables

#### 2.4.1. Anthropometric Variables

Anthropometric parameters at baseline and after interventions included the body mass index (BMI, as the body weight in kg/m^2^), weight loss (WL, as ([Initial BMI]—[Post-intervention BMI])), percent of total weight loss (%TWL, as ([Initial Weight]—[Post-intervention Weight])/([Initial Weight])) × 100, and the percentage excess BMI loss (%EBMIL, as (ΔBMI/[Initial BMI–25]) × 100). We evaluated systolic and diastolic blood pressures by using automatic tensiometer device (Omron M2-HEM-7121-E, Kyoto, Japan). Blood samples were centrifuged for 20 min at 2.500 *g* and the obtained plasma was tested for glucose and lipid determinations. The fasting glucose and glycated hemoglobin (A1C), and the lipid profile (triglycerides, total cholesterol, LDL-cholesterol, and HDL-cholesterol) were quantified by standard methods (ADVIA 2400 Chemistry System, Siemens, Germany) in the Analytical department of Hospital Fundación Jiménez Díaz. All variables were measured before and after interventions.

#### 2.4.2. Analysis of Microbiota

Fecal samples were obtained in OMNIgene-GUT tubes (Abyntek, Spain) at the beginning and after treatments, and stored at −80 °C. Patients did a self-collection at home, following the manufacturer’s instructions. They took fecal samples free from urine or toilet water with a spatula, and transferred them into provided tubes (with homogenizer and stabilizing liquid). Samples were kept for one week at room temperature and delivered to the dietician, who froze them (−80 °C) until use. The OMNIgene-GUT kit provides a valid method to keep RNA at room temperature [22]. After two months, total RNA was extracted from feces (~50 mg) by dissolving in Trizol reagent (Thermo Fisher). RNA concentration and purity were assessed by the 260/280 nm-ratio using the Nanodrop spectrophotometer (Nirko). Equal amounts of RNA were reverse-transcripted to obtain the cDNA for quantitative-PCR (qPCR), as previously described [23]. The gene expression assays were labelled with Fam fluorophore, whereas the housekeeping gene was labelled by VIC fluorophore. Amplification conditions were: 2′ at 50 °C, 10′′ at 95 °C and 40 cycles of 15′′ at 95 °C and 1′ at 60 °C (AB7500 fast y Quant Studio 5; Thermo Fisher). All samples were prepared in triplicate to obtain their threshold cycle (Ct). If deviation for each triplicate were higher than 0.3 cycles, Ct was not considered. The relative expression for each gene was achieved following the model R = 2^−ΔΔCt^. The primer setup was designed to target the ribosomal RNA genes (16S) of the major bacterial groups present in the mammalian intestinal microbiota, including the Firmicutes, Bacteroidetes, Proteobacteria, Fusobacteria, Actinobacteria, and Verrucomicrobia phyla [24]. To gain an insight into the bacterial composition, we used specific primers for bacterial species (Table 2). The specificity of primers was checked in silico with the “probe match” facility of the Ribosomal Database Project (http://rdp.cme.msu.edu/), and further validated on the BLAST search (NCBI) [25]. The primers were purchased from Thermo Scientific, and stored at −20 °C. The reference ranges for intestinal bacteria were calculated as an average of number of gene copies (NGC) from fecal samples of a control population of volunteer patients (Appendix A). These subjects (Spanish; 50% females; 45.0 ± 5.0 years-old; *n* = 100) were non-obese; normoglycemic and normolipidemic; and free from known cardiovascular, malignant, and digestive or intestinal diseases. Thus, characteristics of this control group could be compared with those of the study group.

### 2.5. Statistics

Quantitative variables were summarized as mean values and standard deviation, or by median and interquartile range, depending on the symmetry of the data distribution. Variables with normal distribution were expressed as mean values and standard deviation, whereas those variables with non-normal distribution were shown as median and interquartile ranges. Normality of quantitative variables was analyzed by the Shapiro Wilk test. Variables with normal distribution were compared using Student’s t test for independent and paired samples, while variables with non-normal distribution were compared using the Mann–Whitney U test for independent samples, and a Wilcoxon Signed Rank test for relate samples. Associations between variables were studied by the univariate linear regression or quantile regression. Values of *p* < 0.05 were considered significant. Statistical analyses were performed using the statistical package for social science (SPSS, IBM, Armonk, NY, USA), version 26.0.

## 3. Results

### 3.1. Characterization of the Obese Population with Gut Dysbiosis

A total of twenty patients (14 females, mean age 46.4 ± 5.7 years-old; and 6 males, mean age 41.0 ± 12.1 years-old) with mostly class-I obesity were included in this pilot study (Table 1). At baseline, their body weight and the body mass index (BMI) were 87.8 ± 8.4 kg and 32.2 (5.3) kg/m^2^, respectively. According with the established pathophysiological parameters of blood pressure and glycemia [26,27], they were in the limit of normotension and normoglycemia. Their lipid profile was also slightly altered [28], showing a minor elevation of triglycerides, total cholesterol, and LDL-cholesterol, whereas HDL-cholesterol was in the normal range.

In addition, this obese cohort with a potential low risk of metabolic and cardiovascular disease showed an altered composition and distribution of bacterial gut microbiota. By directed qPCR analysis of fecal samples, we evaluated the presence of muconutritive, immunomodulatory, and proteolytic bacteria. We observed an overall decrease in bacteria number, with a reduction in Firmicutes and mostly Bacteroidetes phylum (8.3 and 7.5 log NGC/g, respectively), compared to reference parameters (Table 2). Among Firmicutes, *Faecalibacterium sp* and *Enterococcus spp* were lessened. In the Bacteroidetes phylum, there was a notorious diminution of *Bacteroides spp*. Thus, the ratio of Firmicutes/Bacteroidetes was 0.5, which was over the reference range. In contrast, these patients showed normal levels of the Proteobacteria and Fusobacteria phyla (Table 2), but a robust decrease in the Actinobacteria and Verrucomicrobia phyla, particularly in the *Bifidobacterium spp* and *Akkermansia muciniphila,* respectively. As expected, these data suggest that obese patients exhibited gut dysbiosis with a significant alteration in the number and distribution of, particularly, muconutritive and immunomodulatory bacteria.

### 3.2. Reduction of the Body Weight and CV Risk Factors by PENS-Diet +/− Probiotics

Since patients displayed an elevated BMI with altered microbiota, we next examined the effect of an anti-obesity strategy based on satiety neurostimulation and intake of a hypocaloric diet (PENS-Diet), with or without administration of multistrain probiotics. Patients were randomly divided into two groups (*n* = 10, each), with no significant differences in age, gender, body weight, BMI, blood pressure, glycemia, and lipid profile (Table 1). Gut microbiota was also akin in both groups (Table 2). Thus, before interventions, anthropomorphic characteristics and microbiota distribution were similar between groups. After ten weeks, patients with a PENS-Diet showed significant reductions in body weight and BMI (Table 3). BMI was reduced by 13% and dropped into the overweight range (28.0 kg/m^2^). PENS-Diet also decreased systolic and diastolic blood pressure, fasting glucose, triglycerides, and total cholesterol. Interestingly, patients with a PENS-Diet + probiotics exhibited a similar effect by ameliorating 20% of body weight and BMI (26.3 kg/m^2^), as well as blood pressure, fasting glucose, A1C, and triglycerides. Moreover, no adverse effects were found in both groups of subjects.

By further comparison between both therapeutic approaches (Table 4), PENS-Diet + probiotics unveiled a significantly higher weight loss (16.2 vs. 11.1 kg, respectively; *p* = 0.022) and total weight loss (%TWL) (17.5 vs. 12.9%, respectively; *p* = 0.02) than the PENS-Diet intervention. The excess BMI lost (%EBMIL) was also significantly higher (84.2 vs. 57.0%, respectively; *p* = 0.021) after probiotics. Moreover, plasma A1C, triglycerides and HDL-cholesterol levels were more reduced (−0.46 vs. −0.05 mg/dL, *p* = 0.032; −47.0 vs. −8.5 mg/dL, *p* = 0.002; and 10.5 vs. 0.05 mg/dL, *p* = 0.005, respectively) (Table 4). These data suggest that an administration of multistrain probiotics to a PENS therapy under hypocaloric diets could further decrease the body weight, glycemia, and dyslipidemia in obese patients. In this regard, we tested the potential associations between probiotics and the body weight parameters, A1C, and lipid levels. By univariate linear regression, probiotics administration was significantly associated with the difference of WL, %TWL, %EBMIL, and A1C. Similarly, by quantile regression, probiotics was associated with the difference of TG and HDL (Figure 1A). Indeed, probiotics showed a positive association with WL, %TWL, %EBMIL, and HDL, while it was negative with A1C and TG (Figure 1B).

### 3.3. Microbiota Modifications after PENS-Diet +/− Probiotics

Alterations in human obesity, glycemia, and lipidemia could parallel changes in gut microbiota [29]. In fact, PENS-Diet showed a tendency to enrich some specific bacteria (i.e., *Prevotella spp*, *Bifidobacterium spp*) and to improve the Firmicutes/Bacteroidetes ratio (Table 5). However, PENS-Diet + probiotics was able to increase *Prevotella spp* (+1.3%, *p* = 0.05) and further reduce the Firmicutes/Bacteroidetes ratio (0.10). This intervention also stimulated *Bifidobacterium spp* (+51.2%; *p* = 0.005) and *Akkermansia muciniphila* (+41.1%, *p* = 0.033) growth (Table 5). Thus, an addition of probiotics to anti-obesity intervention with a PENS-Diet may help to attenuate the altered Firmicutes/Bacteroidetes ratio in an obese gut, and to enrich its content of *Prevotella spp,* and mostly, Actinobacteria *(Bifidobacterium spp)* and Verrucomicrobia *(Akkermansia muciniphila)* bacteria.

## 4. Discussion

The great majority of obese subjects are in the class I obese category, which considerably increases morbidity and public health expenses. In the US Centers for Disease Control and Prevention [30], epidemiological data indicate that approximately 2/3 of obese men and 50% of obese women are in this group. Although the mortality rate within class I obesity is similar to normal weight, the risk of developing T2DM, hypertension, dyslipidemia, metabolic syndrome, obstructive sleep apnea, cancer, and non-alcoholic fatty liver disease is notoriously elevated [31]. Therefore, the evidence calls attention to finding more effective and safe therapies for these patients. In this regard, PENS of dermatome T6 has been proposed as an alternative to pharmacological products and surgical procedures to decrease appetite and weight loss, allowing a better compliance of hypocaloric diets. PENS was initially applied to morbidly obese patients awaiting bariatric surgery, in order to reduce the pre-surgery body weight [10]. Later, we and others extended this technique to patients with overweight and mild-to-moderate obesity. PENS or the hypocaloric diet induced by themselves a significant but slight reduction in body weight (3.6 and 5.6 kg, respectively). However, the combination of both PENS and a diet revealed a mean of weight loss over 10–14 kg, with maintained effects for at least one year after therapy [8,9,32]. Now, in class I obese individuals, we show that ten weeks of PENS-Diet displayed a similar reduction in body weight (11.1 kg) and an improvement of blood pressure and the glycemic and lipid profiles. These effects could be justified by the caloric restriction and by the neurostimulation of the gastric wall and promotion of early satiety [9]. However, an alteration in gut microbiota could have also played a key role. In this regard, the etiopathogenetic of obesity is multifactorial and data from literature suggest a contribution of intestinal dysbiosis in obesity development [33]. Our obese subjects unveiled a microbiota alteration, with a reduction of muconutritive and immunomodulatory bacteria such as *Akkermansia muciniphila, Faecalibacterium sp,* and *Bifidobacterium spp*. In consonance, the Firmicutes/Bacteroidetes ratio was considerably elevated. Importantly, these variations in microbiota may have influenced on obesity development [34,35]. However, PENS-Diet tended to enrich *Prevotella spp* and consequently, the balance of Firmicutes/Bacteroidetes was slightly lessened. The amelioration of this ratio has been frequently linked with an improvement of weight loss and intestinal inflammation and permeabilization [36], and although a precise taxonomic characterization of the bacteria would have been discerned between species, *Prevotella spp* can lead to beneficial effects on mucin regulation, glucose metabolism, and hepatic glycogen storage [37]. Undoubtedly, several lifestyle factors (e.g., smoking, sedentarism, stress, circadian rhythms, personal hygiene, ovarian cycle) may have also altered intestinal microbiota. In this sense, a 20% rate of menopause (without hormonal supplementation) was described in both groups of patients.

In this line, the addition of a multistrain probiotic to the PENS-Diet could have further enhanced these favorable effects. Adomelle^®^ is formulated by *Lactobacillus plantarum LP115, Bifidobacterium brevis B3,* and *Lactobacillus acidophilus LA14*. Supplementation of *L. plantarum* in humans and mice induced a body fat decrease and muscle mass increase, enhancing energy harvest and anti-fatigue effects [38,39]. In obese mice, *L. plantarum* also reduced insulin resistance, plasma triglycerides and proinflammatory factors [40]. *L. acidophilus*, when combined with phenolic compounds or other probiotics, induced weight loss in overweight adults [41]. Moreover, it promoted a significant improvement of glucose homeostasis and cholesterol metabolism in obese mice, by gene downregulation of glucose transporters, cholesterol precursors, and immune factors [42]. Finally, administration of *B. breve-B3* ameliorated the body fat in obese individuals and rats due to its ability to conjugate linoleic acid from diets [43,44]. In our study, the addition of these probiotics to the PENS-Diet further improved the body weight, plasma A1C, triglycerides, and HDL-cholesterol. Moreover, a clear tendency was found for fasting glucose, TC and LDL-cholesterol, which could reach statistical significance in a larger group of patients. Previous randomized controlled trials showed that administration of probiotics alone slightly reduced the body weight and BMI (−0.55 kg and −0.3 kg/m^2^, respectively) in parallel with fasting glucose (−0.35 mg/dL) and lipids (total cholesterol, −0.43 mg/dL and LDL-cholesterol, −0.41 mg/dL) [19]. Particularly, single probiotics such as *Lactobacillus gasseri, Bifidobacterium animalis*, and *Pediococcus pentosaceus* achieved higher benefits than multiple probiotics (i.e., combinations of *Bifidobacterium spp., Lactobacillus spp.*, and/or *Lactococcus spp.*) Thus, a combination of diverse anti-obesity strategies could lead to better outcomes against obesity. In particular, probiotics may induce summative effects when administered together with PENS and a hypocaloric diet. In fact, this triple intervention exhibited a synergic action on reduction of body weight and cardiovascular risk factors. Likely, the promotion of early satiety induced by PENS could be helped by the ingestion of a salutary diet of low caloric intake, and by the balance of healthy microbiota. In this sense, the addition of probiotics further enhanced *Prevotella spp*, *Bifidobacterium spp*, and *Akkermansia muciniphila*, promoting a more muconutritive and immunomodulatory microbiota. *Bifidobacterium spp* have been demonstrated to be positive for the gastrointestinal barrier function and for immunoregulation [45]. By increasing the abundance of *Bifidobacterium spp* (i.e., with prebiotic oligofructose), gut permeability was reduced in obese mice, in correlation with a decrease in LPS and inflammatory markers [46]. In these animals, when combined with *L. acidophilus*, *Bifidobacterium spp* also enriched microbiota composition [47]. Moreover, *Bifidobacterium spp* produced lactate, which is transformed into butyrate by butyrate-producing bacteria in the intestine (i.e., *Prevotella ruminicola*) [48]. These SCFAs play a crucial role in cardiovascular homeostasis and lipid and glucose metabolism by supplying energy and producing glucagon-like peptide-1, peptide YY, and leptin [49]. Furthermore, butyrate induces mucin synthesis and protects intestine integrity by increasing tight junction assembly. In addition, *A. muciniphila* can also regulate gut permeability [50]. Its abundance was inversely correlated with adipose tissue inflammation and insulin resistance in mice and humans [51,52]. In obese hyperlipidemic mice, *A. muciniphila* also improved metabolic endotoxemia, vascular inflammation, and atherosclerotic lesions [53]. Altogether, the enrichment of the muconutritive and immunomodulatory bacteria observed in our patients could also participate in the improvement of their plasma metabolic and cardiovascular factors, and in the attenuation of their body weight.

### Limitations of the Study

In this pilot study, we evaluated the addition of probiotics to an anti-obesity strategy with the PENS-Diet. Probiotics administration further reduced body weight, but its effect on the waist circumference was not evaluated. The abdominal obesity might be affected by microbiota changes, and its quantification would also add key information on the risk of cardiovascular disease. In addition, since multiple factors could influence the effects of probiotics, our data should be taken with care. Unknown comorbidities or habits may alter bacterial distribution and probiotics action. Furthermore, different physical levels and skills could have affected the practice of daily exercise and subsequent weight loss in patients. A personalized control of these practices by an exercise specialist might also improve the adherence and outcomes of this work. Finally, for a group of subjects who follow only a diet regime, PENS intervention or probiotics intake could offer interesting and comparative data about potential changes in microbiota distribution. Therefore, all these variables will be considered in a future study. In this line, the estimation of the sample size per group of obese patients will be at least thirty (two-side significance level; α = 0.05 and power 1 – β = 0.8), following the published formula [54]. According to previous works, the difference of BMI among treatments would be at least 3.0 ± 4.87 kg/m^2^ [55,56].

## 5. Conclusions

A Mediterranean-like hypocaloric diet helped to decrease the body weight, and associated hyperglycemia/hyperlipidemia and blood pressure in class-I obese patients intervened with PENS. However, the addition of *Lactobacillus plantarum LP115*, *Bifidobacterium brevis B3*, and *Lactobacillus acidophilus LA14*) promoted a positive influence on anti-obesogenic gut bacteria by increasing muconutritive (*Akkermansia muciniphila*) and immunomodulatory (*Bifidobacterium spp*) microbiota, and Bacteroidetes phylum (*Prevotella spp*). Consequently, the Firmicutes/Bacteroidetes ratio was further reduced, and these changes could be mediating, at least in part, the further improvement of the body weight and plasma A1C, triglycerides, and HDL-cholesterol. Therefore, a combined strategy of a hypocaloric diet, PENS and probiotics administration may promote summative effects against obesity and related comorbidities (i.e., cardiovascular diseases). Nevertheless, larger studies are needed to analyze the direct actions and interactions of these strategies on gut microbiota.

## Figures and Tables

**Figure 1 ijerph-17-07239-f001:**
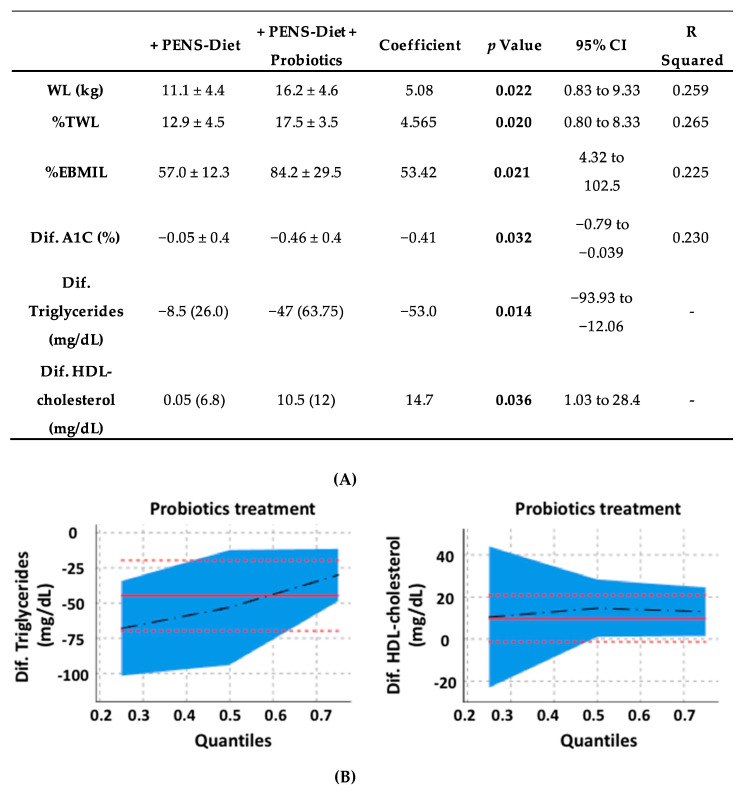
Associations for probiotics and the body weight, glycemia, and dyslipidemia in obese patients. By univariate linear regression, the probiotics administration was significantly associated with WL, %TWL, %EBMIL, and A1C, whereas by a quantile regression, probiotics were associated with TG and HDL (**A**). In bold, the statistically significant data. Probiotics exhibited a positive association with WL, %TWL, %EBMIL, and HDL, while it was negative with A1C and TG (**B**). The associations between variables with a normal distribution were studied by a univariate linear regression, while those with non-normal distribution were studied by a quantile regression. WL, %TWL and %EBMIL, as weight loss, percent of total weight loss, and the percentage excess BMI loss, respectively. A1C and HDL, as glycated hemoglobin, and high-density lipoprotein, respectively.

**Table 1 ijerph-17-07239-t001:** Distribution of age, gender, and baseline blood pressure, glycemic and lipid profile between groups.

	Total Population (*n* = 20)	PENS-Diet (*n* = 10)	PENS-Diet + Probiotics (*n* = 10)	T(df)/U Value	*p* Value
Age (years)	44.7 ± 8.2	45.2 ± 8.9	44.3 ± 7.8	0.24	0.813
Females/Males	14/6	7/3	7/3	-	>0.999
Body weight (kg)	87.8 ± 8.4	84.6 ± 5.1	91.1 ± 10.1	−1.82 (18)	0.085
BMI (kg/m^2^)	32.2 (5.3)	32.2 (2.76)	33.0 (6.82)	48.0	0.912
Systolic blood pressure (mmHg)	137.5 (20.0)	140.0 (20.0)	130.0 (22.5)	41.5	0.529
Diastolic blood pressure (mmHg)	80.0 (8.75)	80.0 (6.25)	80.0 (10.0)	46.0	0.796
Fasting glucose (mg/dL)	95.5 (24.5)	96.5 (29.7)	95.5 (20.2)	48.0	0.912
A1C (%)	5.5 ± 0.7	5.4 ± 0.7	5.6 ± 0.7	−0.61 (18)	0.544
Triglycerides (mg/dL)	148.5 (60.7)	156.0 (96.2)	147.0 (42.5)	49.5	>0.999
Total cholesterol (mg/dL)	199.9 ± 44.0	204.9 ± 52.4	195.0 ± 35.9	0.49 (18)	0.628
LDL-cholesterol (mg/dL)	102.0 (57.0)	114.6 (79.0)	107.0 (44.2)	48.0	0.912
HDL-cholesterol (mg/dL)	45.0 (55.1)	47.7 (22.7)	44.5 (15.0)	38.0	0.393

BMI, A1C, LDL, and HDL, as the body mass index, glycated hemoglobin, and low- and high-density lipoprotein, respectively. T and U values are also shown. Df, as degrees of freedom.

**Table 2 ijerph-17-07239-t002:** Gut microbiota in obese patients.

	Total Population (log NGC/g)	PENS-Diet (*n* = 10)	PENS-Diet + Probiotics (*n* = 10)	T(df)/U Value	*p* Value	Reference Range (log NGC/g)
**Firmicutes phylum**	**8.3 ± 0.86**	8.2 ± 0.94	8.4 ± 0.81	−0.49 (18)	0.630	**8.5–11.0**
*Lactobacillus spp*	4.9 ± 1.13	4.8 ± 1.47	5.0 ± 0.7	−0.5 (18)	0.619	4.5–7.0
*Faecalibacterium sp*	**6.2 ± 1.09**	6.0 ± 1.25	6.5 ± 0.89	−1.04 (18)	0.308	**7.0–9.0**
*Roseburia spp*	6.7 (1.08)	6.6 (1.75)	6.9 (0.85)	33.5	0.218	6.5–8.5
*Bacillus sp*	1.9 ± 0.82	2.0 ± 1.0	1.9 ± 0.67	0.27 (18)	0.979	0–4.0
*Staphylococcus spp*	3.0 ± 0.62	3.1 ± 0.64	2.9 ± 0.62	0.67 (18)	0.509	2.5–5.0
*Veillonella spp*	4.5 ± 0.86	4.4 ± 1.06	4.6 ± 0.63	−0.61 (18)	0.547	4.5–7.0
*Clostridium (Cocc)*	8.2 (1.05)	7.9 (1.7)	8.4 (0.9)	28.5	0.105	7.0–9.0
*Clostridium (Perf)*	3.8 ± 1.19	3.7 ± 1.51	3.9 ± 0.83	−0.23 (18)	0.814	0–5.0
*Enterococcus spp*	**5.9 (1.20)**	5.6 (1.67)	6.2 (1.05)	28.5	0.105	**6.0–8.5**
**Bacteroidetes phylum**	**7.5 (1.63)**	8.1 (1.87)	7.5 (1.0)	40.0	0.481	**8.0–11.0**
*Prevotella spp*	6.7 (3.7)	5.1 (4.1)	7.3 (4.5)	47.5	0.853	5.0–8.5
*Bacteroides spp*	**7.1 ± 1.26**	7.4 ± 1.16	6.7 ± 1.33	1.18 (18)	0.252	**7.5–9.0**
**Firmicutes/Bacteroidetes**	**0.5 ± 0.4**	0.5 ± 0.4	0.46 ± 0.3	0.37 (18)	0.714	**0.1–0.3**
**Proteobacteria phylum**	5.7 (1.89)	6.0 (1.37)	5.0 (1.29)	39.0	0.436	3.0–7.0
*Escherichia coli*	4.5 ± 1,81	4.5 ± 1.79	4.6 ± 1.92	−0.18 (18)	0.859	4.5–7.0
*Pseudomonas spp*	1.0 (1.45)	1.0 (1.42)	1.0 (1.5)	45.5	0.739	0–4.0
*Campylobacter spp*	1.0 (<0.001)	1.0 (<0.001)	1.0 (0.73)	45.0	0.739	0–3.5
*Helicobacter spp*	2.4 (2.1)	1.9 (2.2)	2.4 (1.95)	48.0	0.912	0–3.5
**Fusobacteria phylum**	2.79 ± 1.24	2.42 ± 1.26	3.16 ± 1.15)	−1.36 (18)	0.188	0–4.5
*Fusobacterium nucleatum*	2.79 ± 1.24	2.42 ± 1.26	3.16 ± 1.15)	−1.36 (18)	0.188	0–4.5
**Actinobacteria phylum**	**4.41 ± 2.28**	4.34 ± 2.78	4.48 ± 1.79	−0.14 (18)	0.891	**6.5–9.0**
*Bifidobacterium spp*	**3.85 ± 1.96**	3.8 ± 2.4	3.9 ± 1.6	−0.11 (18)	0.913	**5.5–7.5**
**Verrucomicrobia phylum**	**2.6 (3.65)**	1.8 (2.3)	3.7 (3.47)	30.0	0.143	**5.5–9.0**
*Akkermansia muciniphila*	**2.4 (3.33)**	1.7 (2.1)	3.4 (3.2)	29.0	0.123	**5.0–8.5**

Relevant bacteria phyla; Firmicutes and Bacteriodetes, and its ratio, and Proteobacteria, Fusobacteria, Actinobacteria, and Verrumicrobia, were evaluated in fecal samples before anti-obesity treatments. The reference ranges for the bacteria phyla were obtained from fecal samples of a control population (see Methodology and Appendix A). T and U values are shown. Df, as degrees of freedom. In bold, bacteria levels outside the reference ranges. NGC/g: number of gene copies per gram of feces.

**Table 3 ijerph-17-07239-t003:** Weight loss and improvement of blood pressure, glycemia, and dyslipidemia after PENS-Diet or PENS-Diet + probiotics interventions.

	**Baseline**	**+ PENS-Diet**	**T(df)/W Value**	***p* Value**
Body weight (kg)	**84.6 ± 5.1**	**73.5 ± 3.7**	**7.91 (9)**	**<0.001**
BMI (kg/m^2^)	**32.2 (2.76)**	**28.0 (1.6)**	**0.00**	**0.005**
Systolic blood pressure (mmHg)	**140.0 (20.0)**	**120.0 (2.5)**	**0.00**	**0.018**
Diastolic blood pressure (mmHg)	**80.0 (6.25)**	**70.0 (20.0)**	**7.5**	**0.038**
Fasting glucose (mg/dL)	**96.5 (29.7)**	**88.5 (25.0)**	**0.00**	**0.005**
A1C (%)	5.4 ± 0.7	5.3 ± 0.5	0.36 (9)	0.723
Triglycerides (mg/dL)	**156.0 (96.2)**	**138.5 (80.9)**	**0.00**	**0.005**
Total cholesterol (mg/dL)	**204.9 ± 52.4**	**195.9 ± 46.5**	**1.75 (9)**	**0.004**
LDL-cholesterol (mg/dL)	114.6 (79.0)	132.5 (78.25)	21.0	0.507
HDL-cholesterol (mg/dL)	47.7 (22.7)	51.5 (24.9)	5.0	0.798
	**Baseline**	**+PENS-Diet + probiotics**	**T(df)/W value**	***p* value**
Body weight (kg)	**91.1 ± 10.1**	**74.9 ± 6.7**	**11.09 (9)**	**<0.001**
BMI (kg/m^2^)	**33.0 (6.82)**	**26.3 (4.3)**	**0.00**	**0.005**
Systolic blood pressure (mmHg)	**130.0 (22.5)**	**120.0 (12.5)**	**0.00**	**0.011**
Diastolic blood pressure (mmHg)	**80.0 (10.0)**	**80.0 (15.0)**	**0.00**	**0.041**
Fasting glucose (mg/dL)	**95.5 (20.2)**	**84.0 (11.5)**	**6.00**	**0.028**
A1C (%)	**5.6 ± 0.7**	**5.1 ± 0.4**	3.63 (9)	**0.012**
Triglycerides (mg/dL)	**147.0 (42.5)**	**85.5 (38.7)**	**0.0**	**0.005**
Total cholesterol (mg/dL)	195.0 ± 35.9	176.5 ± 47.2	**1.75 (9)**	0.113
LDL-cholesterol (mg/dL)	107.0 (44.2)	100.0 (46.2)	13.0	0.139
HDL-cholesterol (mg/dL)	44.5 (15.0)	57.0 (20.0)	10	0.074

The body weight, BMI, systolic and diastolic blood pressures, fasting glucose, A1C, and lipid profiles were analyzed after ten weeks of anti-obesity approaches. In bold are the statistically significant data. Variables with normal distribution were compared using Student’s t test for paired samples, whereas variables with non-normal distribution were compared using the Wilcoxon Signed Rank test. T and W values are shown. Df, as degrees of freedom. *p* < 0.05 was considered significant. BMI, A1C, LDL, and HDL, as the body mass index, glycated hemoglobin, and low- and high-density lipoprotein, respectively.

**Table 4 ijerph-17-07239-t004:** Differences in weight loss, blood pressure, and plasma parameters between PENS-Diet and PENS-Diet + probiotics.

	+PENS-Diet	+PENS-Diet + Probiotics	T(df)/U-Value	*p* Value
WL (kg)	**11.1 ± 4.4**	**16.2 ± 4.6**	**2.51 (18)**	**0.022**
%TWL	**12.9 ± 4.5**	**17.5 ± 3.5**	**−2.54 (18)**	**0.020**
%EBMIL	**57.0 ± 12.3**	**84.2 ± 29.5**	**−2.28 (18)**	**0.021**
Dif. Systolic blood pressure (mmHg)	−12.5 (22.5)	−10.0 (12.5)	43.0	0.631
Dif. Diastolic blood pressure (mmHg)	−10.0 (10.0)	−2.5 (10.0)	24.0	0.052
Dif. Fasting glucose (mg/dL)	−7.0 (11.0)	−13.0 (16.5)	31.0	0.165
Dif. A1C (%)	**−0.05** **± 0.4**	**−0.46** **± 0.4**	**2.32 (18)**	**0.032**
Dif. Triglycerides (mg/dL)	**−8.5 (26.0)**	**−47.0 (63.75)**	**11.0**	**0.002**
Dif. Total cholesterol (mg/dL)	−9.0 ± 7.4	−18.5 ± 33.3	0.87 (18)	0.391
Dif. LDL-cholesterol (mg/dL)	0.5 (42.75)	−18.0 (25.5)	26.0	0.075
Dif. HDL-cholesterol (mg/dL)	**0.05 (6.8)**	**10.5 (12)**	**14.00**	**0.005**

The weight loss and percentages of TWL and EBMIL, systolic and diastolic blood pressure, and plasma glucose and lipids (triglycerides, total cholesterol, LDL-c and HDL-c) were compared between both groups of patients. In bold, the statistically significant data. Variables with a normal distribution were compared using Student’s t-test for independent samples, while those with a non-normal distribution were compared using the Mann–Whitney U test. T and U values are also shown. Df, as degrees of freedom. *p* < 0.05 was considered significant. WL, %TWL and %EBMIL, as weight loss, percent of total weight loss, and the percentage excess BMI loss, respectively. A1C, LDL, and HDL, as glycated hemoglobin, and low- and high-density lipoprotein, respectively.

**Table 5 ijerph-17-07239-t005:** Bacterial differences after PENS-Diet or PENS-Diet + probiotics interventions.

	Baseline	+ PENS-Diet	*p* Value	Baseline	+PENS-Diet + Probiotics	*p* Value	T *(df)/U *	*p* *
***Prevotella spp***	5.10 (4.1)	5.25 (2.9)	>0.999	7.30 (4.5)	7.40 (2.5)	0.05	**20.5**	**0.023**
***Bifidobacterium spp***	3.80 ± 2.4	3.90 ± 2.1	0.911	3.90 ± 1.6	5.90 ± 0.9	**0.005**	**−2.27 (18)**	**0.036**
***Akkermansia muciniphila***	1.70 (2.1)	1.50 (2.6)	0.151	3.40 (3.2)	4.80 (1.7)	**0.033**	**13.5**	**0.004**
**Firmicutes/Bacteroidetes**	0.50 ± 0.4	0.40 ± 0.3	0.480	0.46 ± 0.3	0.10 ± 0.5	**0.019**	**2.17 (18)**	**0.043**

*Prevotella spp*, *Bifidobacterium spp*, and *Akkermansia muciniphila* levels (log NGC/g) in obese patients after PENS-Diet or PENS-Diet + probiotics. The ratio of Firmicutes/Bacteroidetes is also shown for both strategies. Variables with a normal distribution were compared using Student’s t test for independent and paired samples, whereas variables with non-normal distribution were compared using the Mann–Whitney U test for independent samples and Wilcoxon Signed Rank test for related samples. *p* < 0.05 was considered significant. T * and U *, as T-value and U-value between PENS-Diet + probiotics and PENS-Diet interventions. Df, as degrees of freedom. *p* *, as *p* value between PENS-Diet + probiotics and PENS-Diet interventions. In bold, the statistically significant data.

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
