# Peer review of "Addition of Probiotics to Anti-Obesity Therapy by Percutaneous Electrical Stimulation of Dermatome T6. A Pilot Study"

_ijerph, 2020, doi:10.3390/ijerph17197239_

Round 1

Reviewer 1 Report

Minor revision requested

Abstract

line 19: please insert "." between the words "factors" and "Objective";

line 30: I would like to suggest to report as "Akkermansia spp, Prevotella spp, ..."

Methodology

line 103: please insert a space after the "...study. Also, ..."

I disagree with the authors' instructions regarding exercise. In my humble opinion, suggesting to different people (therefore with physical, conditional skills and different habits and lifestyle) "identical physical exercise instructions (1h brisk walking / day)..." is not entirely correct. How would the authors define "bisk walking"?  Perhaps this part could be improved by better describing what was suggested to the participants.

In the same way, I am not very convinced by the way in which it has been reported "the adherence rate to diet and exercise", that for the authors "was 100%". Perhaps this aspect could be improved by briefly describing what "a weekly following by dietician" consisted of. Different levels of physical activity practiced could affect the outcome of the work.

Finally, I would suggest that it may be more appropriate to involve an Exercise Specialist to better assess participants' adherence to exercise recommendations. 

Perhaps these limitations could be better highlighted by the authors in "4.1. Limitations of the Study" section.

Author Response

Reviewer #1:

Abstract

line 19: please insert "." between the words "factors" and "Objective";

line 30: I would like to suggest to report as "Akkermansia spp, Prevotella spp, ..."

Thanks for these corrections. We have included the changes (in red)

Methodology

line 103: please insert a space after the "...study. Also, ..."

We have corrected the sentence

I disagree with the authors' instructions regarding exercise. In my humble opinion, suggesting to different people (therefore with physical, conditional skills and different habits and lifestyle) "identical physical exercise instructions (1h brisk walking / day)..." is not entirely correct. How would the authors define "brisk walking"?  Perhaps this part could be improved by better describing what was suggested to the participants.

Yes, it is correct. Different people can show different skills and performances to do physical exercise, influencing in the final outcomes of weight loss and reduction of cardiovascular risk factors. We have included this important appreciation in the limitations of the study, and also, we have better described the recommendation of physical practice given by our dietician. In patients under 50 years-old with class-I obesity, brisk walking consisted in a moderate-intensity exercise of walking to a minimum speed of 100 steps per minute (about 4.8 Km/h). Thanks for this important clarification

In the same way, I am not very convinced by the way in which it has been reported "the adherence rate to diet and exercise", that for the authors "was 100%". Perhaps this aspect could be improved by briefly describing what "a weekly following by dietician" consisted of. Different levels of physical activity practiced could affect the outcome of the work.

Thanks for the observation. We have briefly described in the manuscript the weekly follow-up of patients by the dietician to remind and confirm the daily intake of hypocaloric diet and the physical activity. All participants showed a full adherence to the recommended diet and exercise. In this sense, we previously observed a 98% and 94% diet compliance in patients undergoing PENS or PENS-Diet interventions, respectively [9]. The reduction of appetite induced by PENS and the short length of the study could facilitate this high adherence.

Finally, I would suggest that it may be more appropriate to involve an Exercise Specialist to better assess participants' adherence to exercise recommendations. Perhaps these limitations could be better highlighted by the authors in "4.1. Limitations of the Study" section.

            Yes, this is clearly an improvement for our future research. We are designing a larger study analyzing the addition of probiotics to these anti-obesity strategies, and considering the practice and evaluation of physical activity by a fitness trainer. Following your suggestion, we have included it in the "4.1. Limitations of the Study" section. Thanks

Reviewer 2 Report

The study by Dr. Lorenzo and colleagues shows that addition of probiotic to a PENS+hypocaloric diet intervention results in greater weight loss, decreased levels of glycated hemoglobin and triglycerides and improved obesity-linked gut microbiome dysbiosis.

The study deals with the anti-obesity strategies, that is a hot topic, due to the increasing prevalence of this disease in developed countries. As such, and thanks to the evaluation of the putative effect of probiotics on anti-obesity actions, the manuscript has potential impact and relevance. However, the manuscript suffer from a series of issues that need to be addressed by the authors.

The first one concerns the experimental design of the study. A control group exposed only to probiotics, PENS or hypocaloric diet is lacking, as a result of wich one cannot properly discern the effect of probiotics from those of PENS and hipocaloric diet. Certainly, the 3 treatments have a summative effect; but with the current design and statistical analysis it is difficult to determine the contribution of each one of them to the results. It should be better justified why this design is used and not simply stated as a limitation of the study at the end of the paper.

The introduction section should include the idea that obesity is a multifactorial disease and therefore therapeutic approaches should be diverse as well. It should not be forgotten that reducing food intake through hypocaloric diets or PENS it is important, but other factors as physical activity and behavior therapy are also important.

The quality of the tables and figure are not fully satisfactory. There are errors in the numbering of the tables (table 2 should be 1, and viceversa). The legend in figure 1 seems not to correspond what is represented in it. In table 5, you should try to get all the results on a single line and capitalize “Pens” in the sixth column. Respect to information in Table 5, please provide statistical information showing that there are no baseline differences in gut microbiota between the groups (this is especially important for Prevotella spp and Akkermansia muciniphila levels). We must be absolutely sure that the effect of probiotics is not caused by baseline differences between groups.

In general, in the result section I miss data from the degrees of freedom (df) and the t-value. I think that giving only p-value is not enough.

 Finally, the discussion does not address the fact that obese participant receive a triple intervention. Most of the section has focused only on probiotics as if this were the only treatment. It does not address how the synergistic effect of the three interventions may be occurring. In other words, the probiotics would have the same effects whether they were administered alone or in conjunction only with PENS or hypocaloric diet? In line 309, the authors give a brief explication of this point, but it is not enough (What does it mean that there is a amelioration rate of -0.55 in BW? To compare, it is necessary to know what your amelioration rate in these variables is). Please, develop this idea further. It must be clear what benefits has your proposal of intervention has over existing ones or about other less complex treatments.

Author Response

The study by Dr. Lorenzo and colleagues shows that addition of probiotic to a PENS+hypocaloric diet intervention results in greater weight loss, decreased levels of glycated hemoglobin and triglycerides and improved obesity-linked gut microbiome dysbiosis. The study deals with the anti-obesity strategies, that is a hot topic, due to the increasing prevalence of this disease in developed countries. As such, and thanks to the evaluation of the putative effect of probiotics on anti-obesity actions, the manuscript has potential impact and relevance. However, the manuscript suffer from a series of issues that need to be addressed by the authors.

The first one concerns the experimental design of the study. A control group exposed only to probiotics, PENS or hypocaloric diet is lacking, as a result of which one cannot properly discern the effect of probiotics from those of PENS and hypocaloric diet. Certainly, the 3 treatments have a summative effect; but with the current design and statistical analysis it is difficult to determine the contribution of each one of them to the results. It should be better justified why this design is used and not simply stated as a limitation of the study at the end of the paper.

Thanks for the interesting annotation. In a previous study, we had observed that PENS associated with the hypocaloric diet produced a significantly greater weight loss (BMI= -16%) than only PENS (BMI= -4.6%) or the isolated hypocaloric diet (BMI= -6.4%) [9]. Also, data from literature have shown that single or multistrain probiotics alone produced minimal changes in the body weight (BMI= -0.36 and -0.15 Kg/m2, respectively) and in glycemic/lipidemic factors [19]. Therefore, we have now treated obese subjects, who previously were unsuccessfully treated only with the hypocaloric diet, with PENS or  PENS+probiotics under the same diet in order to observe potential differences in weight loss, associated cardiovascular factors (i.e., blood pressure, glycemia and lipidemia), and microbiota. Nevertheless, we agree that the information of the unique treatments may offer interesting and comparative data about potential changes in weight loss and related factors. Following your recommendation, we have included this justification in the Methodology and have modified the Limitations of the Study section (in red).

The introduction section should include the idea that obesity is a multifactorial disease and therefore therapeutic approaches should be diverse as well. It should not be forgotten that reducing food intake through hypocaloric diets or PENS it is important, but other factors as physical activity and behavior therapy are also important.

            Yes, indeed. Therapeutic approaches against obesity should consider at least the diet, exercise, and the psychological behavior of patients. We have included this comment in the Introduction section, with an appropriated reference. Thanks for this suggestion

The quality of the tables and figure are not fully satisfactory. There are errors in the numbering of the tables (table 2 should be 1, and viceversa). The legend in figure 1 seems not to correspond what is represented in it. In table 5, you should try to get all the results on a single line and capitalize “Pens” in the sixth column.

            Sorry for this inconvenience. We sent the tables and figures in the right order and quality, but were apparently modified in the editing process. The Figure 1 also lost the 1A part. Therefore, we have sent it again to hope they conserve the original format. Thanks

Respect to information in Table 5, please provide statistical information showing that there are no baseline differences in gut microbiota between the groups (this is especially important for Prevotella spp and Akkermansia muciniphila levels). We must be absolutely sure that the effect of probiotics is not caused by baseline differences between groups.

            The required data have been included in Table 2, and referenced in the Results section. Thanks for this suggestion

In general, in the result section I miss data from the degrees of freedom (df) and the t-value. I think that giving only p-value is not enough.

            We have added the corresponding data of degrees of freedom and t-value in the legend of the figures, when applied. Thanks again.

Finally, the discussion does not address the fact that obese participant receive a triple intervention. Most of the section has focused only on probiotics as if this were the only treatment. It does not address how the synergistic effect of the three interventions may be occurring. In other words, the probiotics would have the same effects whether they were administered alone or in conjunction only with PENS or hypocaloric diet? In line 309, the authors give a brief explication of this point, but it is not enough (What does it mean that there is a amelioration rate of -0.55 in BW? To compare, it is necessary to know what your amelioration rate in these variables is). Please, develop this idea further. It must be clear what benefits has your proposal of intervention has over existing ones or about other less complex treatments.

            We agree with your comment and consequently, we have rewritten some paragraphs of the Discussion and Conclusion sections. First, we have included our previous data on PENS and on the hypocaloric diet applied in obese patients, as unique treatments. Both strategies induced by themselves a significant but slight reduction in the body weight (3.6 and 5.6 Kg, respectively). However, the combination of both PENS and diet revealed a mean of weight loss over 10-14 Kg, with maintained effects for at least one year after therapy [8][32][9]. Also, as we mentioned, previous trials showed that administration of probiotics alone slightly reduced the body weight and BMI (-0.55 Kg and -0.3 Kg/m2, respectively) in parallel to fasting glucose (-0.35 mg/dL) and lipids (total cholesterol, -0.43 mg/dL and LDL-cholesterol, -0.41 mg/dL) [19]. Thus, a combination of diverse anti-obesity strategies could lead to better outcomes. In fact, this triple intervention exhibited in our patients a synergic action on reduction of the body weight and cardiovascular risk factors. Likely, the promotion of early satiety induced by PENS could be helped by the ingestion of a salutary diet of low caloric intake, and by the balance of healthy microbiota.

Reviewer 3 Report

For the benefit of the reader, there are, however, still several questions need to be answered and clarified. The following comments and suggestions should be taken into account in order to improve the overall quality and readability of the manuscript.   

Major concern:

  1. English check of the manuscript should be performed by native English-speaking professional.
  2. Why is lack of a group of only consumption of probiotics? How to confirm these effects are owing to probiotics or the additive effect of the two.
  3. The food intake should be described.
  4. Was the food intake changed after probiotics consumption?
  5. Please explain why only BW, HbA1C, and TG were decreased, but not other blood parameters.
  6. Whether other probiotics has also the same effects or not?

Author Response

For the benefit of the reader, there are, however, still several questions need to be answered and clarified. The following comments and suggestions should be taken into account in order to improve the overall quality and readability of the manuscript.   

Major concern:

  1. English check of the manuscript should be performed by native English-speaking professional.

Thanks, the English language has been examined by an English-speaking specialist

  1. Why is lack of a group of only consumption of probiotics? How to confirm these effects are owing to probiotics or the additive effect of the two.

Thanks for the interesting annotation. In a previous study, we had observed that PENS associated with the hypocaloric diet produced a significantly greater weight loss (BMI= -16%) than only PENS (BMI= -4.6%) or the isolated hypocaloric diet (BMI= -6.4%) [9]. Also, data from literature have shown that single or multistrain probiotics alone produced minimal changes in the body weight (BMI= -0.36 and -0.15 Kg/m2, respectively) and in glycemic/lipidemic factors [19]. Therefore, we have now treated obese subjects, who previously were unsuccessfully treated only with the hypocaloric diet, with PENS or  PENS+probiotics under the same diet in order to observe potential differences in weight loss, associated cardiovascular factors (i.e., blood pressure, glycemia and lipidemia), and microbiota. Nevertheless, we agree that the information of the unique treatments may offer interesting and comparative data about potential changes in weight loss and related factors. We have included this justification in the Methodology and have modified the Limitations of the Study section (in red).

  1. The food intake should be described.

We have included a brief description of the hypocaloric diet. Patients were recommended to take skimmed milk (200 mL) or natural yogurt and bread (200 g) as breakfast, 100 g of fruit (e.g., apple, pear) in middle morning, and 200 g of vegetables (e.g., spinach, lettuce, cauliflower) or pasta soup, fish (120 g) or chicken (100 g), and fruit (100 g), at lunch and dinner. Olive oil (30 cc) could be also taken as complement, and skimmed milk (200 mL) with coffee or tea as snack. Thanks for the observation.

  1. Was the food intake changed after probiotics consumption?

The food intake was not altered after probiotics consumption. This has been stated in the Methodology section. Thanks.

  1. Please explain why only BW, HbA1C, and TG were decreased, but not other blood parameters.

The addition of our multistrain probiotics to PENS-Diet further improved the body weight, plasma A1C, triglycerides and HDL-cholesterol. Also, a clear tendency was found for fasting glucose [-13.0 (16.5) vs. -7.0 (11.0); p= 016], TC [-18.5 ± 33.3 vs. -9.0 ± 7.4; p= 0.39] and LDL-cholesterol [0.5 (42.75) vs. -18.0 (25.5); p= 0.07] for PENS-Diet-probiotics and PENS-Diet, respectively, which could reach statistical significance in a larger group of patients. In consequence, we have planned a new larger study to better analyze these and other obesity-related factors. Thanks for the annotation. We have included it in the Discussion section.

  1. Whether other probiotics has also the same effects or not?

Yes, previous randomized controlled trials showed that administration of probiotics alone slightly reduced the body weight and BMI (-0.55 Kg and -0.3 Kg/m2, respectively) in parallel to fasting glucose (-0.35 mg/dL) and lipids (total cholesterol, -0.43 mg/dL); LDL-cholesterol, -0.41 mg/dL) [19]. Particularly, single probiotics such as Lactobacillus gasseri, Bifidobacterium animalis and Pediococcus pentosaceus achieved higher benefits than multiple probiotics (i.e., combinations of Bifidobacterium spp., Lactobacillus spp. and/or Lactococcus spp.). Thus, probiotics including at least some strains of Lactobacillus spp and Bifidobacterium spp like ours, could promote positive actions on microbiota equilibrium and in the attenuation of obesity and related cardiovascular factors. Thanks again   

Reviewer 4 Report

The manuscript titled: “Addition of probiotics to anti-obesity therapy by percutaneous electrical stimulation of dermatome T6. A pilot study” presents valuable information regarding the effectiveness of adding selected commercial prebiotics to a Mediterranean-based diet on obese individuals subjected to percutaneous electrical stimulation. The study was well conducted, the selection of subjects was done correctly, methods and the development are accurate. Results and discussion are very well written and consistent. I only have minor corrections to add to make this paper suitable for publication in the International Journal of Environmental Research and Public Health. 

INTRODUCTION

1. Line 38: Could the authors provide a more updated reference? 

2. Lines 41-44. There are plenty of studies reporting the association of obesity and pro-inflammatory factors and the mentioned non-communicable diseases. Could the authors provide more updated references regarding this link?

METHODOLOGY

3. Table 2 could be titled as “Table 1” and vice versa. 

RESULTS

Figure 1. Please provide a higher quality image. 

4. I would recommend that the authors indicate what statistical test was conducted on each table and figure. 

DISCUSSION

5. Line 265: Could it be “increase” instead of “increased”?

CONCLUSIONS

6. Line 355: Please replace “more large” with “larger”. 

Author Response

The manuscript titled: “Addition of probiotics to anti-obesity therapy by percutaneous electrical stimulation of dermatome T6. A pilot study” presents valuable information regarding the effectiveness of adding selected commercial prebiotics to a Mediterranean-based diet on obese individuals subjected to percutaneous electrical stimulation. The study was well conducted, the selection of subjects was done correctly, methods and the development are accurate. Results and discussion are very well written and consistent. I only have minor corrections to add to make this paper suitable for publication in the International Journal of Environmental Research and Public Health. 

INTRODUCTION

  1. Line 38: Could the authors provide a more updated reference? 

Yes, we have updated the reference (in red). Thanks

  1. Lines 41-44. There are plenty of studies reporting the association of obesity and pro-inflammatory factors and the mentioned non-communicable diseases. Could the authors provide more updated references regarding this link?

Yes, we have updated the paragraph and included new references. Obesity itself is a health risk factor that influences the development and progression of various metabolic and cardiovascular diseases, such as dyslipidemia, type-2 diabetes mellitus (T2D), hypertension, and ischemic heart disease, thereby worsening the quality of life of patients and survival [3]. This non-communicable disease is also associated with low-grade, chronic systemic inflammation by dysregulation of adipokines and pro-inflammatory mediators (i.e., cytokines, chemokine), and subsequent alterations in the immune cell composition and distribution [4]. Thanks for this important annotation

METHODOLOGY

  1. Table 2 could be titled as “Table 1” and vice versa. 

RESULTS

Figure 1. Please provide a higher quality image. 

Sorry for this inconvenience. We sent the tables and figures in the right order and quality, but were apparently modified in the editing process. The Figure 1 also lost the 1A part. Therefore, we have sent it again to hope they conserve the original format. Thanks

  1. I would recommend that the authors indicate what statistical test was conducted on each table and figure. 

Following your suggestion, we have included the statistical test on each table and figure. 

DISCUSSION

  1. Line 265: Could it be “increase” instead of “increased”?

CONCLUSIONS

  1. Line 355: Please replace “more large” with “larger”. 

Yes, sorry about that, we have corrected the words.

Round 2

Reviewer 2 Report

The manuscript has improved considerably since the last version. However, I still have some comments. In addition, some typographical errors need to be corrected:

  • Line 90: includes the symbol of % after value of changes in the body weight
  • According to International System of Units, the correct abbreviation for kilogram(s) is kg and not Kg (for example, line 82, 262, 334, …)

I still have some problems with the results section. First, you repeat in the legend of table 1 and 2 exactly the same phrase that already appears in section 2.5. Please, avoid duplicating information.  On the other side, it is important to keep in mind that the appropriate way to report the results of a t-test is: t(degrees of freedom)=the t statistic, p=p value. My advice is to insert in the text itself the results obtained. Consider the same for analysis using Mann-Whitney U test (U= the U value; p=p value)

Right now, you only give a description of results, without giving evidence of the statistical data. At last, you effectively include analysis of baseline differences in gut microbiota between groups in table 2; however, you need to tell your reader what your analysis tested for. I did not see any mention of theses analysis in the text.

Please, unify the way you refer to the two study groups. Sometimes you name them as group 1 and 2 (for example, in table 1 and 2) and then as +PENS-Diet or +PENS-Diet+probiotics

Author Response

The manuscript has improved considerably since the last version. However, I still have some comments. In addition, some typographical errors need to be corrected:

  • Line 90: includes the symbol of % after value of changes in the body weight

Thanks. In that meta-analysis [19] the BMI data are given as kg/m2. Thus, we have adapted our previous results also as kg/m2 (in red)

  • According to International System of Units, the correct abbreviation for kilogram(s) is kg and not Kg (for example, line 82, 262, 334, …)

We have corrected this error along the work. Thanks

I still have some problems with the results section. First, you repeat in the legend of table 1 and 2 exactly the same phrase that already appears in section 2.5. Please, avoid duplicating information.  

Sorry, we have deleted the duplicate information of the legends. Thanks again

On the other side, it is important to keep in mind that the appropriate way to report the results of a t-test is: t(degrees of freedom)=the t statistic, p=p value. My advice is to insert in the text itself the results obtained. Consider the same for analysis using Mann-Whitney U test (U= the U value; p=p value)

Thanks again, we have inserted these data [T(df) and U values] as a new column in the tables 1, 2, 4, and 5. In table 3 of related-variables, we have showed the T-value(df) and the Wilcoxon Signed Rank (W) value.

Right now, you only give a description of results, without giving evidence of the statistical data. At last, you effectively include analysis of baseline differences in gut microbiota between groups in table 2; however, you need to tell your reader what your analysis tested for. I did not see any mention of theses analysis in the text.

Thanks for the suggestion. We have mentioned in the text that “Thus, before interventions, anthropomorphic characteristic and microbiota distribution were similar between groups”.

Please, unify the way you refer to the two study groups. Sometimes you name them as group 1 and 2 (for example, in table 1 and 2) and then as +PENS-Diet or +PENS-Diet+probiotics

Ok, it has been changed. Thanks

Reviewer 3 Report

The revised manuscript is now suitable for publication in this journal. 

Author Response

Thanks

This manuscript is a resubmission of an earlier submission. The following is a list of the peer review reports and author responses from that submission.

Round 1

Reviewer 1 Report

This paper showed the results of a prospective and randomized pilot study aimed to test whether ten weeks of addition of probiotics (Adomelle®, composed by Lactobacillus plantarum LP115, Bifidobacterium brevis B3 and Lactobacillus acidophilus LA14) could improve weight loss and cardiovascular risk factors in 20 obese subjects after Percutaneous electrical stimulation (PENS) and  hypocaloric diet (a 1,200 Kcal/day diet followed a Mediterranean-style of carbohydrates 51%, proteins 23% and fat 26%). This is a prospective and randomized pilot study involving 14 females and 6 males (44.7 ± 8.2 years-old) allocated to PENS T6 + hypocaloric diet  group (n = 10 subjects) and PENS T6 with hypocaloric diet and the addition of probiotics (n = 10 subjects).

Before and after interventions, the authors measured: Anthropometric variables (body mass index, weight loss, percent of total weight loss and the percentage excess BMI loss), blood pressure variables (systolic and diastolic blood pressure values), fasting glucose and glycated haemoglobin and the lipid profile (triglycerides, total cholesterol, LDL-cholesterol, and HDL-cholesterol). Furthermore, authors studied  intestinal bacteria through fecal samples (using colony-forming unit). The reference ranges for intestinal bacteria were calculated from fecal samples of a reference population of control  patients (n=100 Spanish people, non-obese, normoglycemic and normolipidemic). Authors concluded that “In obese patients, the addition of specific probiotics to a PENS intervention under hypocaloric diet, further improved weight loss and glycemic and lipid profile in parallel to the amelioration of gut dysbiosis”.

The main, interesting, question seems to be original and the authors give an advance in current knowledge about research topic already well-studied in authors's previous works.

Ethics aspect, as well as methodological and statistical aspects, are well explained. However, in my opinion, some minor changes are required which may be helpful in improving the quality of the paper.

For all reasons, above expressed, I recommend this paper be accept after minor revision.

Minor revision

Methodology Section

- In 2.3 authors reported that “a record of food intake was applied at the beginning and at the end of the study”. Did you have measured some nutritional specific variables ( e.g. adherence to the Mediterranean diet score)? If so, please provide details.

- In 2.3 authors reported that “all patients received identical physical exercise instructions (one hour of brisk walking a day)”: did you use any counselling protocol? If so, please provide details on the protocol used in the methods section.

-Did you have measured some exercise variables? If so, please provide details.

- In 2.4.3 authors reported that “quantitative variables were summarized as mean values and standard deviation, or by median and interquartile range, depending on the symmetry of the data distribution”. Please clarify it in the tables note.

Results Section

-If you have measured some exercise variables, I would suggest reporting them in the text or in a table.

-In 3.1 authors reported: “A total of twenty patients with mostly class-I obesity were included in this pilot study. They were mainly women with a mean age of 44.7 ± 8.2 years-old (Table 1)”. I would suggest to report: “A total of twenty patients (14 females, mean age… ± … and 6 males, mean age…± …)”.

Discussions Section

- Authors reported that “the amelioration of the Firmicutes/Bacteroidetes ratio has been frequently linked with an improvement of weight loss and intestinal inflammation and permeabilization [33], and although a precise taxonomic characterization of the bacteria would have being able to discern between species, Prevotella spp have shown beneficial effects on mucin regulation, glucose metabolism, and hepatic glycogen storage [34]. In this sense, several lifestyle factors could obviously alter intestinal microbiota, and the hypocaloric diet (with high vegetable intake) may be at least in part responsible for this modulation of microbiota”. Please clarify the important role of lifestyle factors, even though it’s not its primary focus.

- I fully agree with what the authors reported in the "limits" section: “despite our patients previously had unsuccessfully tried the hypocaloric diet as treatment against obesity, a group of subjects who would have followed a diet regime only, could add interesting information in relation to its potential microbiota changes”. Obese patients have many difficulties in adhering to nutritional advice and treatments: for this reason I would suggest to the authors to supplement the text with more information about their patients' adherence to nutritional treatment/advice.

Table 3, table 4 and table 5

I would suggest to report p value legend (using *) instead “in bold” character.

Reviewer 2 Report

In this paper, the authors compare the efficacy of multistrain probiotic in the treatment of obesity on a small sample of Spanish individuals with obesity. Interestingly, the authors enhanced the treatment by percutaneous electrical stimulation of dermatome T6.

Major issues:
A) did all participants finish the study? How did the authors assume to perform intention-to-treat or per-protocol analysis?
B) I am not too fond of very long articles. I know that this might be frustrating for authors because they did their best to enrich the text. However, the introduction and discussion are just boring and could be written in a better style.
C) What was the name of IRB which analyzed the protocol and approved the study? Please, provide nr of consent.
D) We need more information about the control group. How was the group collected? Could the authors compare the characteristics of the study group and the control group?
E) statistical analysis is very plain. Firstly, the study needs a primary endpoint. I may suggest e.g., % of groups achieve a decrease in BMI of more than 5% or 2.5%. The authors should use both univariate and multivariate linear regression models. The dependent variable should be related to the primary aim – the difference in BMI. Alternatively, the authors may use logistic regression models. To compare changes over time using repeated-measures ANOVA.
F) Because this is a pilot study, it is worth calculating the sample size of the study group in future trials.
G) How the authors registered adverse effects?
H) Why some reference ranges are expressed as ranges, and some have one side cut-off?
I) The linear regression models with one categorical independent variable are very poor – the figure with the regression slope should be removed.
J) Here are many statistical comparisons. The authors should use a method for adjusting p-value to multiple comparisons.

Minor issues:
a) abstract should have less than 200 words
b) where is available the dataset?
c) Many fragments of the discussion can be moved to the introduction or removed.
d) references should be written as "[1,2]", not "[1][2]"
e) tables are not edited in the journal style
f) no abbreviations are described under the tables
g) use the term "total population" not "global population"
h) p-value equal to 0.05 is not statistically significant according to the authors' criteria

Reviewer 3 Report

The authors found addition of probiotics to PENS-diet treatment enhanced the effects of anti-obesity. The authors investigated gut microbiota changes through qPCR. The qPCR results should be shown for each patient and methods should be well documented because qPCR results may easily change depending on the protocol.

For qPCR protocol, the authors should provide primer sequence and annealing temperature. Plus, melting curve results as supporting data. 

Omnigene Gut tubes may not be suitable keeping RNA at room temperature.

Gut microbiota may be biased with RT-qPCR. (Just qPCR may be more acceptable, but recent trend tells they should do MiSeq)

Followings are the specific comments

Abstract
dyslipemida => dyslipidemia
"and" should not be italicized

Introduction
"low bacterial richness" => "low bacterial species richness" or "low bacterial diversity"
Is it ok to change "investigate the effect of probiotics on anti-obesity effects of PENS in conjunction with a hypoaloric diet." ?

Method
an hypocaloric => a hypocaloric
as previously => as previously described
I am not sure if Omnigene GUT tubes can store RNA for a week at room temperature.
For bacteria, I think it is 16S only not 23S
Primer sequences used in this study should be given
The authors expressed quantitative values of bacteria in CFU, but the method used in the study was RT-qPCR. CFU is a number of colonies, while RT-qPCR tells you activities of bacteria. In other words, RT-qPCR results vary according to the bacterial condition or environment. I suggest, the authors just use number of gene copy instead.

Results

"As expected [25]" it is strange to have citation here

"(not shown)" I think this is the key result in the study, thus should be shown

Table 2: this should show "before" and "after" the treatment